# Correlates of Spirituality among African American and Black Caribbean Emerging Adults

Meredith O. Hope [1,*], Ann W. Nguyen [2], Robert Joseph Taylor [3] and Linda M. Chatters [3,4]

1. Department of Psychology, The College of Wooster, Wooster, OH 44692, USA
2. Jack, Joseph and Morton Mandel School of Applied Social Sciences, Case Western University, Cleveland, OH 44106, USA; nguyena@case.edu
3. School of Social Work, University of Michigan, Ann Arbor, MI 48109, USA; rjtaylor@umich.edu (R.J.T.); chatters@umich.edu (L.M.C.)
4. School of Public Health, University of Michigan, Ann Arbor, MI 48109, USA
* Correspondence: mhope@wooster.edu

**Abstract:** Spirituality is a significant cultural strength and resource for Black emerging adults. Numerous studies show that increasing numbers of emerging adults tend to identify themselves as being spiritual but not religious. However, no studies to date have identified the demographic correlates of spirituality for Black emerging adults from different ethnic groups (i.e., African American and Black Caribbean). Data from the National Survey of American Life were used to identify the demographic correlates of two indicators of spirituality (e.g., self-rated spirituality and subjective spirituality) among African American and Black Caribbean emerging adults using linear regression. For African Americans, being a woman predicted both greater self-rated spirituality and subjective spirituality. However, among Black Caribbeans, men rated spirituality higher in importance than women. African Americans who reported higher educational attainment tended to report higher levels of self-rated spirituality and subjective spirituality. Romantic status for Black Caribbeans, whether they had no romantic involvement or a current romantic partner, was associated with lower self-ratings of spirituality. Unmarried cohabiting individuals in both ethnic groups tended to report lower levels of self-related spirituality. Unemployment for Black Caribbeans was associated with lower subjective spirituality. Findings are of interest to those who serve and work with Black emerging adults.

**Keywords:** spirituality; religion; Black Americans; emerging adulthood; emerging adults

## 1. Introduction

An abundance of research highlights the important role of spirituality in the lives of individuals of all ages. Over the last two decades, scholarship has grown significantly regarding the transition from adolescence to adulthood, a period known as emerging adulthood (defined as persons 18–29 years of age). However, because little is known about spirituality among this group, research is needed to understand spirituality for individuals within this critical developmental period. Emerging adulthood provides numerous opportunities for individuals to explore new ways of being and knowing as they reassess previously held beliefs, values, and traditions from their earlier years (Arnett 2000; Arnett 2007). During this period of life, emerging adults are more likely to move away from organizational forms of religious involvement (Cooper and Mitra 2018), with some emerging adults favoring more individualized means to seek out and experience the sacred, however they choose to define it (Koenig 2015).

Spirituality reflects the individual's search for, connection to, and experience with what they believe to be sacred (Mattis and Jagers 2001). Spirituality may be informed by religious traditions, beliefs, and values. Conversely, it may also be informed by other sources that lack explicit connections to a particular belief system. Importantly, those

who identify as spiritual may not profess or endorse beliefs in God or a higher power (e.g., nature spirituality). Ultimately, spirituality can offer individuals opportunities to craft their own unique path (Weddle-West et al. 2013). A small but growing corpus of literature explores spirituality among emerging adults. However, very little is known about spirituality and its correlates for Black Americans during the period of emerging adulthood (see Voisin et al. 2016). This study aims to identify demographic correlates of spirituality for Black emerging adults in the United States.

### 1.1. Spirituality and Its Role for Black Americans

Spirituality is often defined as personalized searches for and understanding of the sacred. Although scholars generally agree that spirituality provides personalized pathways for meaning-making, its distinctions from religion remain unclear (Mattis and Jagers 2001). For example, some Black individuals report that they are both religious and spiritual, indicating that they consider religion and spirituality to be both connected and distinct (Boyd-Franklin 2010). In contrast, others use "religious" and "spiritual" interchangeably (Arrey et al. 2016). The interchangeability of these terms can also be seen in how some researchers assess beliefs, values, and practices. Dill (2017) conceptualized prayer as a form of spiritual coping, whereas Chatters et al. (2008a) consider prayer as religious coping. We do not purport to know which conclusion is "correct". Rather, we suggest that these diverse perspectives demonstrate the 'real-world' nuance associated with the study of how individuals make sense of their lives and their world. Reflective of this, the primary focus of this paper is to explicate the role of spirituality as a distinct construct that may play a significant role in Black lives.

Historically and contemporaneously, spirituality has played a major role in the lives of Black Americans. Spirituality functions as a significant psychological resource and a critical source of resilience for this population. For example, sermonic traditions based on liberation and defiance theology provide a spiritual framework for coping with adversity, social, political, and economic marginalization, and racism (Frazier and Lincoln 1974). Moreover, Black churches, a central fixture in Black spirituality, have played an important role in addressing the distinctive needs and concerns of Black Americans and are linked to the history and struggles of this group (Pew Research Center 2021). In fact, close to 30% of Black adults surveyed by the Pew Research Center believe that Black churches have done a great deal to help Black Americans move toward equality in the U.S. (Pew Research Center 2021).

Black Americans are the most religious and spiritual group in the U.S., endorsing the highest rates of religious and spiritual beliefs and practices compared to other racial groups (Chatters et al. 2009; Pew Research Center 2015). While religion and spirituality tend to decline in early adulthood (Ingersoll-Dayton et al. 2002), Black emerging adults, nonetheless, attribute great importance to spirituality (Pew Research Center 2021). Spirituality is a critical psychological and coping resource for Black emerging adults, who are likely to experience a number of major and potentially stressful life changes during this period (e.g., moving out of the family home, starting college, developing serious romantic relationships). Qualitative research reveals that Black emerging adults conceptualize spirituality both as a personal relationship with a higher being or God and as spiritual beliefs and practices that are grounded within traditional religious ideology and practices (e.g., service attendance) (Voisin et al. 2016). This is consistent with national survey data demonstrating that the vast majority of Black young adults (96%) say that they believe in a higher power or God, and close to half of all Black young adults say that religion is important to them (Pew Research Center 2021). Regarding spiritual practices, nearly one out of five Black emerging adults meditate daily, and approximately two-thirds of all Black emerging adults believe in the healing power of prayer (Pew Research Center 2021). Further, nearly half of all Black emerging adults indicate that they rely on prayer and personal religious reflection when they make major life decisions (Pew Research Center 2021). Together, these findings demonstrate that spirituality is both central and important to Black emerging adults.

Although research documents spirituality's long-standing and multifaceted role in Black American experiences, this scholarship primarily focuses on Black adults and older adults. Black adults generally report high levels of spirituality. Furthermore, important historical, demographic, and contextual factors have had an impact on the meaning and functions of spirituality in their lives. For example, as demographic factors, such as older age, female gender, and residence in the South, have been consistently associated with higher levels of religious involvement, these factors may also influence spirituality (Taylor et al. 2009). Black ethnicity (i.e., African Americans and Black Caribbeans), a virtually unexplored topic, may also influence spirituality in distinctive ways. For example, prior work found that African American and Black Caribbean adults both reported similarly high levels of self-rated spirituality; however, African American adults were more likely to perceive themselves as very spiritual (Taylor et al. 2009).

*1.2. Black Emerging Adults and Spirituality*

Similar to studies regarding the role of spirituality for Black adults in the United States, Black emerging adults report that spirituality is important to them (Dill 2017; Jagers and Smith 1996; Voisin et al. 2016). Contextual and demographic factors may also affect the level of spirituality reported by Black emerging adults. For example, (Weddle-West et al. 2013) observed that African American college-attending emerging adults at predominantly white institutions tended to report higher levels of spirituality than their counterparts at historically Black institutions. They suggested that contextual factors (e.g., being Black at a historically Black institution) might provide other sources of support and validation that do not necessarily exist at non-HBCUs. Regarding gender, some studies suggest that Black emerging adult women experience spirituality differently than their male counterparts (Walker and Dixon 2002), while others indicate that Black emerging adult women and men are equally spiritual (Berkel et al. 2004; Dennis et al. 2005).

Black emerging adults report experiencing multiple dimensions and functions of spirituality. African American emerging adult men identify spirituality as an integral factor in shaping their identity (Dancy II 2010). African American emerging adult women reported they harnessed spirituality to navigate their daily lives in predominantly white environments and that spirituality helped them cope, make sense of uncertainty, and receive direction and hope (Patton and McClure 2009). African American emerging adult men connected with their spirituality through the practice of prayer, especially as a conduit for responding to stress (Riggins et al. 2008). In a more recent study, Black gay and bisexual emerging adult men reported that playing and listening to music, whether in solitude or for an audience, were powerful factors in their spirituality (Means 2017).

Black emerging adults conceptualize spirituality in diverse ways. In one critical study, Voisin et al. (2016) explored the multifaceted role of spirituality for African American emerging adults. Results from semi-structured interviews suggested that Black emerging adults shared diverse definitions of spirituality. First, spirituality can encompass their relationship with God or a higher power. Second, spirituality included their reliance on God or the divine to provide guidance and meaning in all facets of life. Third, they associated spirituality with organizational and non-organizational forms of religious behaviors and practices, such as attending worship services, praying, or helping others. Similarly, in a later study, Black LBGQ undergraduates defined spirituality as their relationship with a higher power, asserting that spirituality can be further understood as "internal congruence" and as reflected in their relationships with others (Means et al. 2018). This shows that beliefs and concepts from religious frameworks can provide conduits for spirituality. Moreover, faith-informed spirituality may indicate a transformative process, one that tailors the organized elements of religiosity into intra-personal experiences, perspectives, and understanding. Fourth, emerging adults understood spirituality through secular frameworks. Lastly, conceptualizations of spirituality may vary by gender. For example, African American emerging adult men tended to endorse more secularized perspectives regarding spirituality in their lives. Taken together, it is clear that spirituality cannot be monolithically defined.

Conceptualizations of spirituality may also emphasize practices through which individuals connect with their spirituality (Taylor and Chatters 2010; Benson 2004). A qualitative study using semi-structured interviews with Black emerging adults (Dill 2017) explored how spirituality informs strategies for dealing with adverse life situations. One Black emerging adult man shared that he prays and puts on his "spiritual jacket" for protection. Other emerging adults who identified as either Christian or Muslim identified three ways that they expressed their spirituality: talking about their faith, transferring worries and concerns to "God's hands", and helping the less fortunate in their communities (Dill 2017). Finally, in another study, spirituality was assessed by asking African American undergraduates to rate how influential spiritual beliefs were in their lives (Walker and Dixon 2002). Given the lack of empirical consensus on what constitutes spirituality within Black emerging adult samples, identifying potential correlates of spirituality may elucidate our understanding of how, when, and which aspects of spirituality matter for Black emerging adult development.

*1.3. Focus of the Current Study*

Our study investigates the correlates of spirituality among a national sample of African American and Black Caribbean emerging adults. Most scholarship on emerging adult spirituality tends to reflect the experiences of white emerging adults. The few studies that document the role of spirituality for Black emerging adults are comprised of convenience samples within university contexts. Further, the limited amount of research on demographic correlates of spirituality among Black Americans tends to focus on young, middle-aged and older adults. Although emerging adults are often included in adult samples, they are typically not identified as such, nor are their specific experiences within this developmental stage considered.

To our knowledge, this is the first study to investigate demographic correlates of spirituality for emerging adults from two Black ethnic groups—African Americans and Black Caribbeans. There is limited research on spirituality among African Americans and almost no research among Black Caribbeans in the United States. In addition to addressing this gap in the literature, this study's several advantages over previous research include the use of a national probability sample, examining two measures of spirituality, using a measure of marital/romantic status that reflects the life course of emerging adults, and investigating these issues among both African Americans and Black Caribbeans.

The current analysis used data from the National Survey of American Life: Coping with Stress in the 21st Century (NSAL). Although the NSAL data are over 20 years old, they remain the only available dataset with (1) detailed measures of spirituality and (2) large, nationally representative samples of African American and Black Caribbean adults. To the best of our knowledge, there are no other datasets that have a sizeable and nationally representative sample of Black Caribbeans. Except for Hispanic ethnicity, most datasets do not distinguish between ethnicities among Black Americans.

## 2. Method

The National Survey of American Life: Coping with Stress in the 21st Century (NSAL) was collected by the Program for Research on Black Americans at the University of Michigan's Institute for Social Research. The fieldwork for the study was completed by the Institute of Social Research's Survey Research Center, in cooperation with the Program for Research on Black Americans. A total of 6082 interviews were conducted with persons aged 18 or older, including 3570 African Americans, 891 non-Hispanic whites, and 1621 Blacks of Caribbean descent. For the purposes of this study, Black Caribbeans are defined as persons who trace their ethnic heritage to a Caribbean country but who now reside in the United States, are racially classified as Black, and who are English-speaking (but may also speak another language). The overall response rate was 72.3%. The response rates for individual subgroups were 70.7% for African Americans, 77.7% for Black Caribbeans, and 69.7% for non-Hispanic Whites. For the purposes of this paper, the age range of emerging adults is

18–29. Among this population, 806 were African American, and 436 were Black Caribbeans, for a total of 1242 persons of 18–29 years of age. This emerging adult sub-sample is used in this study. Respondents were compensated for their time. The data collection was conducted from 2001 to 2003 (see Jackson et al. 2004a, 2004b for a more detailed discussion of the NSAL sample).

### 2.1. Sample

African American and Black Caribbean emerging adults are distinctive from one another in several respects (Table 1). The African American sample is much more likely to be female (56.19%) than the Black Caribbean sample (43.61%). Black Caribbean emerging adults have much higher levels of family income and are more likely to cohabit and be in a current romantic relationship than their African American counterparts. Consistent with the geographical distributions of these populations, Black Caribbean emerging adults are more likely to reside in the Northeast (54.51%), and African American emerging adults are more likely to reside in the South (58.94%). Black Caribbeans are also much more likely to have been born in another country. African American and Black Caribbean emerging adults were similar with regard to years of education and labor market participation.

**Table 1.** Demographic Characteristics of the Sample and Distribution of Study Variables.

| | Race/Ethnic Groups | |
| --- | --- | --- |
| | **African American** | **Black Caribbean** |
| Self-Rated Spirituality, Mean (SD) | 3.61 (0.61) | 3.56 (0.27) |
| Very Spiritual % (n) | 29.20 (238) | 31.57 (135) |
| Fairly Spiritual % (n) | 51.60 (416) | 51.89 (216) |
| Not Too Spiritual % (n) | 16.00 (127) | 10.05 (56) |
| Not Spiritual at All % (n) | 3.21 (23) | 6.49 (25) |
| Importance of Spirituality Mean (SD) | 3.07 (0.70) | 3.09 (0.32) |
| Very Important % (n) | 70.13 (573) | 66.52 (285) |
| Fairly Important % (n) | 21.69 (176) | 25.04 (105) |
| Not Too Important % (n) | 7.36 (49) | 6.83 (30) |
| Not Important at All % (n) | 0.83 (6) | 1.60 (13) |
| Age Mean (SD) | 22.98 (3.21) | 23.58 (1.42) |
| Gender, % (n) | | |
| Male | 43.81 (275) | 56.39 (189) |
| Female | 56.19 (531) | 43.61 (247) |
| Years of Education, Mean (SD) | 12.40 (1.73) | 12.97 (0.86) |
| Family Income, Mean (SD) | 31,953.34 (31,729.33) | 48,200.41 (18,467.60) |
| Labor Market Participation, % (n) | | |
| Employed | 71.70 (581) | 76.26 (311) |
| Unemployed | 16.84 (135) | 12.61 (69) |
| Not In Labor Force | 11.46 (90) | 11.14 (56) |
| Romantic/ Marital Status, % (n) | | |
| Married | 12.76 (103) | 14.32 (63) |
| Cohabit | 11.96 (86) | 19.38 (50) |
| Current Relationship | 37.27 (315) | 46.10 (185) |
| No Relationship | 38.01 (300) | 20.21 (137) |
| Region, % (n) | | |
| Northeast | 15.88 (83) | 54.51 (301) |
| North Central | 18.57 (133) | 4.41 (4) |
| South | 58.94 (548) | 26.67 (127) |
| West | 6.61 (42) | 14.41 (4) |
| Foreign Born, % (n) | | |
| Born in the United States | 96.99 (778) | 53.61 (202) |
| Born in another country | 3.01 (22) | 46.39 (229) |

Percents and N are presented for categorical variables and Means and Standard Deviations are presented for continuous variables. Percentages are weighted and frequencies are un-weighted.

*2.2. Measures*

Two dependent variables in this analysis assessed spirituality: self-rated spirituality and the self-reports of the importance of spirituality. Self-rated spirituality was measured by the question: "How spiritual would you say you are?" The response categories were as follows: very, fairly, not too, or not at all. The importance of spirituality was measured by the question, "How important is spirituality in your life?" The response categories were as follows: very important, fairly important, not too important, or not important at all.

Several sociodemographic factors (i.e., ethnicity, age, gender, family income, education, labor market participation, marital/romantic status, and region), which have known associations with religion and spirituality, are included. Ethnicity is coded African American and Black Caribbean. Age and education are coded in years, and family income is coded in dollars. Labor market participation differentiates respondents who are employed (the reference category), unemployed, and out of the labor force (e.g., full-time students, disabled). Region is coded into four categories (Northeast, North Central, West, and South). For Black Caribbean respondents only, we included the variable of foreign-born (born in the United States or born in another country).

Our marital/romantic status variable was created to be more appropriate for emerging adults because of the smaller number of this population who are widowed or divorced. We created a composite measure based on marital status and two measures of relationship status. Marital status was assessed using a single item that asked respondents if they were currently married, living with a partner, separated, divorced, widowed, or never married. All currently unmarried, non-cohabiting respondents were additionally asked whether they were currently involved in a romantic relationship. Respondents who indicated that they were not involved in a romantic relationship were then asked, "Do you want a main romantic involvement?" The responses to these questions were combined to construct a four-category measure of romantic involvement: (1) currently married, (2) currently cohabiting, (3) has a romantic relationship, and (4) does not have a romantic relationship. This variable assesses respondents' current relationship status, and all response categories are mutually exclusive.

*2.3. Analysis Strategy*

Ordinary least squares regression was utilized to examine the correlates of spirituality. Computations for the distribution of the sociodemographic characteristics and linear regression analyses were conducted using SAS 9.1.3, which uses the Taylor expansion approximation technique for calculating the complex design-based estimates of variance. All analyses utilize sampling weights. Weights in the NSAL data account for unequal probabilities of selection, non-response, and post-stratification such that respondents are weighted to their numbers and proportions in the full population. SAS uses the Taylor expansion technique for calculating the complex design-based estimates of variance. This corrects standard error estimates in analysis using complex sample designs (i.e., clustering and stratification).

**3. Results**

Information about the levels of spirituality among respondents in our sample is presented in Table 1. Eight out of ten African American and Black Caribbean emerging adults indicate that they are either very spiritual or fairly spiritual, and 7 out of 10 respondents report that spirituality is very important to them. Self-rated spirituality is reported by a lower percentage of both African American and Black Caribbean emerging adults than the importance of spirituality in everyday life. This same pattern is evident among Black adults across the lifespan (Taylor et al. 2009).

Regression analysis testing ethnic differences in spirituality among Black emerging adults is presented in Table 2. There were no significant differences between African American (reference category) and Black Caribbean emerging adults in either self-rated spirituality or the importance of spirituality.

**Table 2.** Ethnic differences among Black Emerging Adults in Self-Rated and the Importance of Spirituality.

|  | Self-Rated Spirituality | | Importance of Spirituality | |
|---|---|---|---|---|
|  | b | S.E. | b | S.E. |
| African Americans Excluded Category | 0 | 0 | 0 | 0 |
| Black Caribbeans | −0.07 | 0.11 | −0.08 | 0.14 |

b = regression coefficient, S.E. = standard error. All regressions control for age, gender, income, education, marital/romantic status, labor marker participation, region and foreign born.

Regression analysis of the demographic correlates of spirituality among African Americans and Black Caribbean emerging adults is presented in Table 3. Among African American emerging adults, gender and education are significantly associated with both measures of spirituality. African American women report higher levels of both self-rated spirituality and the importance of spirituality than men, and those with more years of education report higher levels of both measures of spirituality. In addition, African American emerging adults in a cohabiting relationship report lower levels of self-rated spirituality than married African Americans.

**Table 3.** Regression analysis of the demographic correlates of spirituality among African American and Black Caribbean emerging adults.

| Independent Variables [a,b] | African Americans | | Black Caribbeans | |
|---|---|---|---|---|
|  | Self-Rated Spirituality b (S.E.) | Importance of Spirituality b (S.E.) | Self-Rated Spirituality b (S.E.) | Importance of Spirituality b (S.E.) |
| Age | 0.01 (0.01) | 0.02 (0.01) | −0.01 (0.01) | 0.02 (0.01) |
| Gender |  |  |  |  |
| Male | 0 | 0 | 0 | 0 |
| Female | 0.13 (0.05) ** | 0.16 (0.06) * | −0.13 (0.13) | −0.47 (0.13) *** |
| Marital/Romantic Status |  |  |  |  |
| Married | 0 | 0 | 0 | 0 |
| Cohabitation | −0.39 (0.12) ** | −0.14 (0.11) | −0.78 (0.22) ** | −0.65 (0.32) |
| Have Current Romantic Involvement | −0.12 (0.08) | 0.03 (0.09) | −0.29 (0.09) ** | −0.34 (0.21) |
| No Current Romantic Involvement | −0.16 (0.08) | 0.15 (0.09) | −0.32 (0.10) ** | −0.25 (0.22) |
| Education | 0.04 (0.01) *** | 0.07 (0.01) *** | 0.03 (0.02) | 0.02 (0.03) |
| Family Income | 0.00 (0.00) | 0.00 (0.00) | −0.01 (0.01) | −0.02 (0.01) |
| Labor Market Participation |  |  |  |  |
| Employed | 0 | 0 | 0 | 0 |
| Unemployed | −0.05 (0.07) | 0.09 (0.09) | −0.21 (0.13) | −0.20 (0.08) * |
| Not in Labor Force | 0.05 (0.07) | 0.06 (0.09) | 0.11 (0.07) | 0.10 (0.12) |
| Region |  |  |  |  |
| South | 0 | 0 | -- | -- |
| Northeast | −0.17 (0.13) | −0.06 (0.07) | -- | -- |
| North Central | −0.11 (0.08) | −0.00 (0.11) | -- | -- |
| West | 0.02 (0.07) | −0.15 (0.13) | -- | -- |
| Foreign Born |  |  |  |  |
| Born in the United States | -- | -- | 0.05 (0.07) | −0.00 (0.11) |
| Born in Another Country | -- | -- | 0 | 0 |
| $R^2$ | 0.08 | 0.07 | 0.17 | 0.19 |
| F | 5.48 *** | 5.30 *** | 5.37 *** | 3.31 ** |
| N | 802 | 802 | 429 | 428 |

b = regression coefficient; S.E. = standard error. Note: Significance test of the individual parameter estimates was based on a complex design-corrected *t*-test. a—Regression coefficients and standard errors are reported. b—Several independent variables are represented by dummy variables. Gender, 1 = female, 0 = male; Marital/Romantic Status, Married is the excluded category; Labor Market Participation, Employed is the comparison category; Region, South is the comparison category; Foreign-Born; 1 = Born in the United States; 2 = Born in Another Country. * $p < 0.05$ ** $p < 0.01$ *** $p < 0.001$.

The correlates of spirituality for Black Caribbean emerging adults differ from those for African Americans. Gender is unrelated to self-rated spirituality among Black Caribbean

emerging adults. However, gender is significantly associated with the importance of spirituality, indicating that men report higher levels than women. Black Caribbeans who are unemployed report that spirituality is less important to them than employed Black Caribbeans. Lastly, married Black Caribbean emerging adults report higher levels of self-rated spirituality than those who are cohabiting, in a current romantic relationship, and those not in a romantic relationship.

## 4. Discussion

The primary purpose of this study was to identify significant demographic correlates of spirituality for Black emerging adults in the United States. We found that African American and Black Caribbean emerging adults were comparable in identifying spirituality as being an important aspect of their lives and identity. The groups, however, differed in the overall pattern of correlates associated with spirituality. The remainder of this section focuses on notable findings regarding gender, marital/relationship status, education, and labor market participation.

There were no significant differences between African American and Black Caribbean emerging adults for the importance of or self-ratings of spirituality. This is consistent with previous research indicating that despite the many differences between African Americans and Black Caribbeans, there were no significant differences in spirituality (Taylor et al. 2009), as well as various measures of religious participation (Chatters et al. 2009).

African American women and spirituality. Previous research confirms consistent gender differences in both spirituality and religiosity within the general U.S. population, as well as African Americans. As a matter of fact, gender most consistently accounts for demographic differences, with women reporting significantly higher levels of spirituality and religiosity. Among African Americans, women report significantly higher levels of self-reported spirituality, the importance of spirituality, being both spiritual and religious, frequency of attending religious services, reading religious materials, and self-rated religiosity (Taylor et al. 2009; Taylor et al. 2014).

Narratives of African American experiences within the United States historically center African American women as significant figureheads of religiosity and spirituality within their families and communities. Historic figures such as Harriet Tubman (1822–1913), Sojourner Truth (died 1883), Mahalia Jackson (1911–1972), and Jarena Lee (1783–1864) are noted for their spiritual connections to God, which upheld their resilience and resistance within anti-Black and misogynistic societal contexts (Lee 1849; Patterson 2013; Truth 1998; Weisenfeld and Newman 2014; Wilson 2019). Spirituality remains a mainstay for contemporary African American women of all ages, with recent scholarship showing that it often serves as a source of strength and inspiration for African American emerging adult women (Leath et al. 2022; Patton and McClure 2009). As such, our finding that African American emerging adult women reported higher levels of spirituality than their male counterparts is consistent with previous empirical literature on gender differences and historical accounts of the centrality of spirituality for African American women.

Black Caribbean men and spirituality. Contrary to previous research on gender differences in spirituality, Black Caribbean emerging adult men reported higher levels of spirituality than women in our study. This was a major departure that is inconsistent with other research on spirituality and religious participation among the general U.S. population and African Americans. Further, research on Black Caribbeans across the entire adult age range found that, compared to their male counterparts, women had higher levels of religious involvement in terms of service attendance, reading religious materials, prayer, and endorsing the importance of taking children to religious services (Taylor et al. 2010). However, research focusing specifically on spirituality among Black Caribbean adults did not find any significant gender differences (Taylor et al. 2009). Collectively, this prior information, coupled with the current findings, suggests that the meaning of spirituality for Black Caribbean men may be different than for African American and white American counterparts. That is, Black Caribbean men's spirituality may not be as linked to their

religious participation as it is for other groups. This may be particularly the case for Black Caribbean emerging adult men. An alternative explanation is that emerging adult Black Caribbean men, in particular, appear to be distinctive in their higher endorsement of the importance of spirituality as compared to their female counterparts. In interpreting these findings, it is important to acknowledge that there is very little research on religion and spirituality among African American and Black Caribbean emerging adults, as well as very little research on these emerging adult populations overall. As such, our discussion of these findings is speculative.

The finding that Black Caribbean men report higher levels of the Importance of spirituality in their lives may have a parallel in previous research in the area of mental health. Research on gender differences in psychiatric disorders generally finds that women report higher rates of depression and suicide attempts as compared to men. This, however, is not the case among Black Caribbeans, where men had higher rates of suicide attempts than Black Caribbean women, African American women, and African American men (Joe et al. 2006). Further, the rate of major depression for Black Caribbean men is similar to that for Black Caribbean women (Williams et al. 2007). Collectively, this information may indicate that the life circumstances of Black Caribbean men are different from those of men of other race and ethnic groups, which is reflected in different levels of spirituality as well as mental health problems. More research is needed to understand the unique life circumstances of Black Caribbean men and how they are different with regard to spirituality and mental health but similar with regard to religious participation.

Romantic relationships and spirituality. Romantic relationship contexts are associated with the importance of spirituality and self-ratings of spirituality for African Americans. Within our study, African American emerging adults who were in cohabitating relationships (as compared to married persons) tended to report lower levels of spirituality. For many African Americans, spirituality is closely intertwined with religious values, traditions, beliefs, and practices (Chatters et al. 2008b). African Americans with past or current exposure to religious teachings and settings may not separate spirituality from religion. Many religious traditions and communities caution congregants and adherents of the faith to reserve sexual intimacy for marriage. For African Americans who have been socialized in these settings and traditions, cohabitation without a societally or religiously recognized marriage may be linked to their lower levels of self-rated spirituality. Lower levels of self-rated spirituality among cohabitors were also seen among Black Caribbean emerging adults. This finding is consistent with previous research among Black Caribbean adults (Taylor et al. 2009). Although cohabitation has gained in frequency and acceptance in the past 40 years, stigma remains associated with this living arrangement, especially among highly religious individuals.

Similarly, research on marital status differences in religion and spirituality generally indicates that married respondents have higher levels of spirituality and religiosity. Consistent with this previous literature, Black Caribbean emerging adults who were married tended to report higher levels of spirituality than Black Caribbean emerging adults who had a current romantic relationship, those who were not involved in a romantic relationship, and those who were unmarried and cohabitating. Many married Black Caribbeans, like other married couples, attend religious services and participate in church-based activities more frequently than their unmarried counterparts (Taylor et al. 2014). This may lead to higher levels of both religiosity and spirituality. Additionally, there may be a self-selection process for both African Americans and Black Caribbeans. That is, young adults who are more religious and spiritual may have a higher likelihood of marrying than other marital/romantic status groups.

Educational attainment and spirituality. Consistent with previous studies that document significant and positive associations between educational attainment and spirituality for African American adults (Taylor et al. 2009), our study found a similar result for African American emerging adults. Among African Americans across the adult age range, years of formal education are associated with more frequent service attendance and participation in

church activities (Taylor et al. 2014). However, among Black Caribbeans, education was not associated with spirituality in previous work across the adult age range (Taylor et al. 2014) nor in our study of emerging adults.

Unemployment and spirituality. We found that Black Caribbean emerging adults who were currently unemployed had lower levels of spirituality. Unfortunately, no previous research examines the relationship between spirituality and employment status among either Black Caribbeans or African Americans. Because employment status can often be linked to other aspects of basic survival (i.e., food, shelter, transportation), individuals who are unemployed may be more focused on securing resources for those needs and be less inclined to view spirituality as being an important part of their lives. The connections between employment status and spirituality are clearly a needed area for future research.

## 5. Implications and Limitations

This study has several notable strengths. This is the first study to identify sociodemographic correlates of spirituality among Black emerging adults. Studies on this topic often focus on adults across the lifespan (aged 18 and older) or older adults, resulting in a dearth of knowledge regarding spirituality and its correlates among emerging adults. Emerging adulthood is an important developmental period that is marked by several major transitions that can be daunting and stressful (e.g., starting college, starting a career, moving out of the family home). Focused research on this developmental period is critical, as spirituality often functions as an important psychological and coping resource for many emerging adults who are facing major life transitions. Another strength of the current analysis is the focus on both the African American and Black Caribbean populations. Research on Black Americans tends to treat the Black diaspora in the U.S. as a monolithic population. However, there are key cultural and sociodemographic differences between various Black ethnic groups in the U.S. By distinguishing between African American and Black Caribbean emerging adults, our analysis provided a more nuanced understanding of factors that contribute to spirituality within the Black diaspora in the U.S. Related to this, a further strength of this investigation is the use of a nationally representative sample of African American and Black Caribbean emerging adults, which provides information on within-group diversity and generalization of study findings to the population level.

Findings should be interpreted within the context of study limitations. First, the data were cross-sectional, and we were unable to determine temporal ordering. Future research should use prospective longitudinal study designs to determine these causal relationships. Second, while our study distinguished between two different Black American ethnic groups—African American and Black Caribbean—there are other Black ethnic groups (e.g., Black immigrants from African nations). Future investigations should explore how spirituality and its correlates may vary across the Black diaspora. Third, the NSAL was collected in 2001–2003, which clearly limits its generalizability. The degree to which the age of our data impacts the generalizability of our results is unknown. Research has shown a decline in religious participation among all groups in the United States over the past 20 years. However, for many people, the decline in religious participation, in particular things like service attendance and church membership, has not resulted in a similar decline in spirituality. For instance, recent data from the Pew Research Center suggest that seven out of ten U.S. adults think of themselves as spiritual people or report that spirituality is very important in their lives (Pew Research Center 2023). Despite these important limitations, the current study has many significant contributions by providing novel information regarding the correlates of two indicators of spirituality among both African American and Black Caribbean emerging adults.

**Author Contributions:** Conceptualization, M.O.H. and R.J.T.; formal analysis, M.O.H. and R.J.T.; data writing—original draft preparation, M.O.H. and R.J.T.; writing—review and editing, M.O.H., R.J.T., A.W.N. and L.M.C. All authors have read and agreed to the published version of the manuscript.

**Funding:** The preparation of this article was supported by grants from the National Institute on Aging to A.W.N. [(P30AG072959; U24AG058556)], the National Institute on Aging to R.J.T. [(P30 AG015281)] and the National Institute of Diabetes and Digestive and Kidney Diseases to L.M.C. [(P30DK092926)].

**Institutional Review Board Statement:** An IRB Statement is not relevant to this research as it involves secondary data analysis. The data collection itself was approved by the University of Michigan Institutional Review Board.

**Informed Consent Statement:** Informed consent was obtained from all subjects involved in the original study.

**Data Availability Statement:** Data from the National Survey of American Life can be downloaded from the Program for Research on Black Americans (https://www.icpsr.umich.edu/icpsrweb/ICPSR/studies/27121). Accessed on 1 June 2023.

**Conflicts of Interest:** The authors declare no conflicts of interest.

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
