# Peer review of "Correlates of Spirituality among African American and Black Caribbean Emerging Adults"

_religions, doi:10.3390/rel15030341_

Round 1

Reviewer 1 Report

Comments and Suggestions for Authors

I really enjoyed this essay. My focus is on the concept of "spirituality," and the authors are savvy enough to recognize how flexible and variable it can be. They asked their research subjects to give their own definitions, and sure enough, for these persons, "spirituality" might better be labelled as the personal dimension of religion. They link it more closely to God, church worship and prayer than do those in other populations for whom "spirituality" and "religion" function almost as opposites. For both groups in this research study, that is not the case. Where this becomes clear is that  the potentially negative restrictive aspects of religion (such as "No sex outside marriage") directly carry over into "spirituality" here. Thus, low spirituality ratings from those cohabitating. This would absolutely not be the case in a study of white Wiccans, for example. For them, only "religion" is full of such rules, while "spirituality" floats free. 

Beyond this, "spirituality" works to eliminate the negative aspects of "religion." It is impossible to say anything bad about spirituality. Frankly, this may make the concept popular, but also problematic. (Like "growth"- it becomes a glow-word) But as this interesting essay reveals, the two terms, i.e. spirituality and religion, are and have remained bonded together in the minds of some populations,  Thus,,the exact meaning of spirituality is impossible to pin down as a one-size-fits-all concept. 

I recommend publication of this essay, but believe other reviewers are needed who are experts in the 2 populations covered in the essay.

Author Response

We very much appreciate this insightful perspective regarding our manuscript.

Reviewer 2 Report

Comments and Suggestions for Authors

Using a unique data set, the authors examine correlates of self-reported spirituality among a large sample of African American and Black Caribbean emerging adults. Below are some concerns and suggestions, in no particular order.

1. The authors should more clearly justify the study of self-reported spirituality, as a distinct concept and variable apart from conventional religiosity, like belief, prayer, service attendance, etc. Does it capture something that religion variables don't? Is there evidence that it's associated with positive outcomes related to health? The authors routinely discuss spirituality in the context of religion, or as a product of religiosity, but do not adequately make the case for why it should be studies on its own. 

2. On page 4, the authors omit an important piece of the Voisin et al. (2016) article. Many of their interviewees discussed spirituality in the absence of religiosity, rather in the context of responsibility to self and community (and also noted gender differences here). I found it be curious that the authors omitted these findings from their discussion of that research.

3. The NSAL was a really unique and interesting study that probably deserves a bit more explanation and background than the authors give it here. It scope and focus, sample size, sampling method, and more. In the discussion, the authors note potential issues with generalizability due to it being from 2001 to 2003. It would be more accurate to ask whether its findings would still hold up today, particularly in light of declines in religiosity for younger Americans, even including African Americans.

4. Why did the analyses omit measures of religiosity entirely? This seems odd, given the fact that such variables would likely be strongly associated with spirituality (and probably explain the gender gap). And in light of the Voisin et al. (2016) paper, it would be interesting to see just how strongly they're correlated with spirituality, given that it doesn't seem to be universal for African Americans.

5. It would be worthwhile to clarify the direction of the categories in the ordinal variables capturing spirituality, to help with interpreting the results in Table 1. Or even better, it could be worth having a table or bar graph showing the full categories, and percentages. 

Author Response

Thank you for this insightful feedback. We believe that these suggestions have strengthened this manuscript. Please see our responses to each point below. 

  1. We have added a paragraph at the beginning of the Spirituality and its role for Black Americans section to address the importance of studying spirituality by itself, rather than linking it to religiosity.
  2. In response to this insight, we have expanded information about the Voisin study to amplify the importance of studying spirituality.
  3. This is an excellent point. The NSAL was collected in 2001-2003 which clearly limits its generalizability.  It is unknown the degree to which the age of our data impacts the generalizability of our results.  Research has shown a decline in religious participation among all groups in the United States over the past 20 years.  However, for many people the decline in religious participation, in particular things like service attendance and church membership, has not resulted in a similar decline in spirituality.  For instance, recent data from the Pew Research Center finds that seven out of 10 U.S. adults think of themselves as spiritual people or report that spirituality is very important in their lives (Pew, 2023).  Based upon your comment we have addressed this issue in our limitation section.  Pew Research Center, December, 2023, “Spirituality Among Americans”
  4. We understand and appreciate your point. However, adding religion variables to our analysis would distract from the main purpose of our paper.  Previous research by Chatters, Taylor and others have found that among African Americans, Black Caribbeans and Non-Latino whites that spirituality and religion are highly associated.  For some people these two constructs are highly related, whereas for others spirituality and religion are very distinct. For this paper, we wanted to focus solely on the demographic correlates of our two measures of spirituality.  The NSAL is the only data set in which we can perform this analysis.  We see our paper as a companion to the qualitative analysis of Voisin (2016).
  5. This is a very good point. We have revised Table 1 based upon your recommendation.  We have added full categories of our dependent variables and their corresponding percentages. In the introduction, we have added a paragraph that explains the need to examine spirituality separately.

Reviewer 3 Report

Comments and Suggestions for Authors

This is an important original research on emerging adultlhood, which is a critical developmental period in the growth of any individual and is fraught with socio-psychological challenges. Spirituality, therefore, is a crucial source of stemming such challenges, which is why the focus of the research on how spirituality impinges on the life of emerging adults in the African American and Black Caribbean communities is an important unit of analysis.

What is more, identifying the two groups for cross-analysis is a positive departure from the usual monolithic treatment of ethnic groups of African descent.

While the study is well defined and carried out by means of a sound methodological approach, I wonder whether the NSAL has no current data rather than the 2001-2003 data the study employs? Surely, two decades can have a significant impact on human behaviour, particularly in the current rapid changes characteristic of our techno-rational times? The author(s) may well wish to address this question to address a possible accusation of an apparent use of dated data.

Author Response

This is an excellent point and is similar to that raised by Reviewer #2 in their 3rd point.  We note in our response to Reviewer #2 that research (e.g., Pew Research Center, 2023) has shown a decline in religious participation among all groups in the United States in the past 20 years.  However, for many people the decline in religious participation witnessed has not resulted in a similar decline in spirituality. 

Reviewer 4 Report

Comments and Suggestions for Authors

The article is clearly written. It addresses an important research topic that has not been treated and even needs further research. It clearly shows where their research stands in relation to previous research and highlights some new findings on Black Carribean men. It points to diversity among African Americans which is an area in need of further research. It clearly explicates the strengths and limits of the study and draws lines for further research. 

Author Response

Thank you for this comment. We very much appreciate your time and thoughtfulness in reviewing our manuscript.